# TMT-Based Quantitative Proteomic Analysis Reveals the Key Role of Cell Proliferation and Apoptosis in Intestine Regeneration of *Apostichopus japonicus*

**DOI:** 10.3390/ijms25084250

**Published:** 2024-04-11

**Authors:** Chuili Zeng, Ke Xiao, Qilin Shi, Xu Zhan, Chenghua Li

**Affiliations:** 1State Key Laboratory for Managing Biotic and Chemical Threats to the Quality and Safety of Agro-Products, Ningbo University, Ningbo 315211, China; 18243007672@163.com (C.Z.); xiaoke010227@163.com (K.X.); 15659802139@163.com (Q.S.); z15757862257@163.com (X.Z.); 2Laboratory for Marine Fisheries Science and Food Production Processes, Qingdao National Laboratory for Marine Science and Technology, Qingdao 266071, China

**Keywords:** *Apostichopus japonicus*, intestine regeneration, proteomics, TMT

## Abstract

Sea cucumbers are widely known for their powerful regenerative abilities, which allow them to regenerate a complete digestive tract within a relatively short time following injury or autotomy. Recently, even though the histological changes and cellular events in the processes of intestinal regeneration have been extensively studied, the molecular machinery behind this faculty remains unclear. In this study, tandem mass tag (TMT)-based quantitation was utilized to investigate protein abundance changes during the process of intestine regeneration. Approximately 538, 445, 397, 1012, and 966 differential proteins (DEPs) were detected (*p* < 0.05) between the normal and 2, 7, 12, 20, and 28 dpe stages, respectively. These DEPs also mainly focus on pathways of cell proliferation and apoptosis, which were further validated by 5-Ethynyl-2′-deoxyuridine (EdU) or Tunel-based flow cytometry assay. These findings provide a reference for a comprehensive understanding of the regulatory mechanisms of various stages of intestinal regeneration and provide a foundation for subsequent research on changes in cell fate in echinoderms.

## 1. Introduction

Regeneration is an important regulatory phenomenon in metazoa, with broad biological significance, exhibiting a heterogeneous distribution from the lowest to the highest phyla [1]. This phenomenon involves limited flipping or recovery abilities at the cellular and tissue scales, a broader ability to replace missing parts after self-injury or traumatic amputation, and even the ability to reconstruct the entire individual from parts of the body. For example, among widely studied vertebrates and invertebrates, zebrafish can regenerate fins, salamanders can regenerate appendages, and even planaria can regenerate the digestive tract from arbitrary slices [2,3,4]. However, most of the current research on regeneration is external, probably limited by the study of visualization or fluoroscopy of separate individuals, and there is less research on visceral organs. However, research on the regeneration of biological organs such as the intestine is of great significance for the exploration of regenerative medicine [5].

Echinoderms (Echinodermata) are grouped in the deuterate evolutionary branch of animals and, as such, are phylogenetically closely related to vertebrates [6]. Echinoderms are well known for being able to regenerate body parts and thus provide excellent models for studying regenerative processes in adult organisms [7]. The regeneration of a complete digestive system is one of the most dramatic regenerative processes that can be observed in deuterostome animals. This process occurs in echinoderms of the class Holothuroidea, or sea cucumbers, following natural or artificial stimuli. Unlike most of the other species of mammals which possess limited repair ability of the digestive tract, sea cucumbers possess an extraordinary capacity to regenerate their digestive tract after evisceration in the absence of stem cells, and some species even could achieve this in less than one month post evisceration [8,9]. Thus, the intestinal regeneration of sea cucumbers provides a new model to explore the mechanism of internal organs or be used to further develop treatment options for human diseases [10]. In addition, many species of sea cucumbers are important commercial and economical marine animals with nutritional and medical uses, but the intestinal regeneration in sea cucumbers is an additional energy expenditure, which diverts nutrients away from growth and development and exerts a negative effect on the body weight and reproduction [11,12]. Therefore, the intestinal regeneration of sea cucumbers may be a potential pathway not only for the exploration of internal organ regeneration but also for the optimization of its commercial and subsistence fisheries [12].

Intestinal regeneration is a complex process involving multiple cellular activities such as cell dedifferentiation, cell migration, cell proliferation, and cell apoptosis. It cannot be revealed by the expression pattern of a single gene alone. Therefore, using high-throughput sequencing technology to obtain more information on genes or proteins is of great significance for promoting the deep development of intestinal regeneration research [13,14]. With the development of high-throughput sequencing technology, a large-scale gene expression profile related to the regeneration process of sea cucumbers has been constructed, enabling the study of this mechanism at the global transcriptome and proteome levels. The transcriptome of *Apostichopus japonicus* (*A. japonicus*) during intestinal regeneration has been well studied, while proteomic analysis only reported at the normal intestine and the early stage of regeneration (3 days post-evisceration (dpe)) [15,16]. Given that the process of intestinal regeneration is divided into multiple stages, and each process is crucial for regenerating a complete intestine and involves a series of cellular events, we investigated the differential proteins of the entire process of intestinal regeneration using tandem mass tag (TMT)-based proteomics. The comparative study of the responsive profiles in different regeneration stages was investigated by TMT-based proteomics to enhance our understanding of the biological responses to intestine regeneration in echinoderms.

## 2. Results

### 2.1. Overview of Changes in Differential Proteins (DEPs) between Normal and Different Regenerative Stages

The proteomics data derived from the regeneration tissues from the normal group and five regeneration groups (2, 7, 12, 20, and 28 dpe) were used to investigate the underlying molecular pathways during the process of intestine regeneration. The raw data has been submitted to the iProX platform (PXD050025). The result showed that 538, 445, 397, 1012, and 966 DEPs were detected (*p* < 0.05) between the normal and 2, 7, 12, 20, and 28 dpe stages, respectively (Figure 1). The detailed protein expression is shown in Table 1. Compared to the normal group, the top 5 proteins with the highest upregulation at 2 dpe group were Fibrinogen-like protein A (FGLA), Subtilisin-like protease 1 (SUB1), Solute Carrier Family 6 Member 14 (SLC6A14), Ficolin-2 (FCN-2), Tripartite motif-containing 71 (TRIM71), while the top 5 proteins with the highest downregulation were Fatty aldehyde dehydrogenase (FALDH), Angiotensin-converting enzyme (ACE), Thymidylate synthase (TS), Fatty acid-binding protein 2 (FABPI), Enoyl-CoA delta isomerase 1 (ECI1), respectively. At the 7 dpe group, the significantly upregulated proteins were SUB1, Caspase-6 (CASP6), FGLA, Heme-binding protein 2 (HEBP2), Uronyl 2-sulfotransferase-like isoform X3 (UST), while the most downregulated proteins were FALDH, ECI1, TS, FABPI, long-chain specific acyl-CoA dehydrogenase (ACADL). At the 12 dpe group, the top five proteins with the highest upregulation were HEBP2, Mucin-2 (MUC2), Transmembrane prolyl 4-hydroxylase (P4HTM), Exosome component 10 (EXOSC10), Dimethylaniline monooxygenase (FMO), while the most downregulated proteins were FALDH, ACE, ECI1, FABPI, and TS. At the 20 dpe group, the top five proteins with the highest upregulation were SUB1, FGLA, sulfotransferase family cytosolic 1B member 1 (SULT1B1), UST, HEBP2, while the most downregulated proteins were FALDH, ACE, ECI1, FABPI, TS. At the 28 dpe group, the top five proteins with the highest upregulation were Elongation of very long chain fatty acids protein 6 (Elov6), Peroxisomal carnitine O-octanoyltransferase (CROT), Tryptophan-tRNA ligase (TrpS), Transducin beta-like protein 3 (TBL3), Acyl-CoA-binding protein (ACBP), while the most downregulated proteins were ECI1, ACE, FALDH, FABPI and Cysteine-rich motor neuron 1 (CRIM1).

### 2.2. Kyoto Encyclopedia of Genes and Genomes (KEGG) Pathway Enrichment Analysis

KEGG pathway enrichment analysis was performed for the differentially expressed proteins. The upregulated proteins in the normal vs. 2 dpe comparison group were mapped to glycosaminoglycan degradation, whereas the downregulated proteins were mapped to protein processing in retinol metabolism and peroxisome (Figure 2A,B). The upregulated proteins in the normal vs. 7 dpe comparison group were mapped to cysteine and methionine metabolism, while the downregulated proteins were mapped to ubiquinone and other terpenoid-quinone biosynthesis (Figure 2C,D). It is worth noting that the upregulated proteins in the normal vs. 12 dpe group and normal vs. 20 dpe comparison group were mapped to cysteine and methionine metabolism, and the downregulated proteins were mapped to glutathione metabolism (Figure 2E–H). In addition, the upregulated proteins in the normal vs. 28 dpe comparison group were mapped to DNA replication, and the downregulated proteins were mapped to extracellular matrix (ECM) receptor interaction (Figure 2I,G).

### 2.3. Gene Ontology (GO) Analysis of DEPs with Different Expression Trends

To explore the biological molecular functions that could play important roles during the process of intestine regeneration, cluster hierarchy and GO terms enrichment analysis were applied. The results showed that the DEPs were divided into 6 clusters (Figure 3), and the corresponding cluster cellular component (CC), biological process (BP), and molecular function (MF) were analyzed, as shown in Figure 4. It is worth noting that the protein in cluster 1 and cluster 2 presented a gradual upregulation trend, and the expression profiles were mainly in the primary metabolic process, organic substance metabolic process, and organic cyclic compound binding, heterocyclic compound binding, respectively. Cluster 3 showed a downregulation trend throughout the entire regeneration stage, and the expression profiles were mainly related to the metabolic process and catalytic activity. Similarly, cluster 4 showed a downregulation trend from the 0 dpe to 20 dpe and then an upregulation trend in the 20–28 dpe group. The expression profiles were mainly in single-organism transport and single-organism metabolic processes. Cluster 5 was upregulated firstly at 2 dpe and then gradually downregulated to 28 dpe, in which the expression profiles were mainly in protein binding. Finally, the DEPs of cluster 6 were upregulated firstly from the beginning of regeneration to 12 dpe and then downregulated, and the expression profiles were mainly in the metabolic process.

### 2.4. Expression Trends of Proteins Related to Different Fate States during Intestinal Regeneration Process

The process of intestinal regeneration involves the orderly occurrence of a series of cytological events, such as cell migration, cell dedifferentiation, cell proliferation, apoptosis, and cell differentiation [17]. Cell apoptosis and cell proliferation are two important states that oppose and interact with each other [18]. In order to investigate the regulatory role of these two cell fate states in intestinal regeneration, we analyzed the expression of these cell-related proteins from a proteomic perspective, hoping to address the transition of cell fate states from a proteomic perspective. The results were shown in Figure 5A, in which the expression trend of cell proliferation-related proteins was at a low level at both the normal and 2 dpe group, while the expression levels of these proteins gradually increased as intestinal regeneration progressed, reaching a peak at 28 dpe group. We performed EdU-based flow cytometry to verify these results and detect cell proliferation during intestinal regeneration (Figure 5B). The statistical results showed (Figure 5C) in the early stage (2 dpe) of intestinal regeneration, the increment of cell proliferation was not statistically significant (4.35 ± 0.68%) compared with the normal group (3.23 ± 0.2%), while a significant up-regulation of 12% occurred from the 7-dpe onwards, and reached a peak of 65% at 28 dpe group. The results of EdU-based flow cytometry were in agreement with that analyzed by the proteome. In addition, we randomly selected genes (*Cyclin-dependent kinase 6* and *DNA replication licensing factor mcm 5*) related to cell proliferation from Figure 5A and used Quantitative Real-time PCR (qRT-PCR) to detect the mRNA expression changes of these genes during the intestine regeneration process of *A. japonicus*. The results showed that the mRNA expression trend of these genes was consistent with the results of the proteome, showing a gradually upregulated trend (Figure 5D). Similarly, we screened DEPs related to cell apoptosis from the proteome and found that the expression levels of these proteins were not significantly upregulated at normal and 2 dpe levels but showed a trend of first increasing and then decreasing as intestinal regeneration progressed (Figure 5E). Then, we used the tunnel-based flow cytometry to detect apoptosis (Figure 5F). The proteome results are shown in Figure 5G. In the early stage (2 dpe) of intestinal regeneration, the increment of cell proliferation was significant (7.83 ± 0.41%) compared with the normal group (2.79 ± 0.28%). At stages 7 and 12 dpe, the apoptosis rate reached 15.43 ± 0.95% and 26.57 ± 0.67%, respectively. Then, as regeneration progressed, the apoptosis rate was downregulated. The value of the apoptosis rate reached 17.6 ± 1.44% and 9.14 ± 0.93% at the stage 20 dpe and 28 dpe, respectively. In addition, we randomly selected cell apoptosis-related genes (*Caspase 6* and *Caspase 8*) from Figure 5E and used qRT-PCR to detect the mRNA expression changes of these genes during the intestine regeneration process in *A. japonicus*. The results showed that the mRNA expression trend of these genes was consistent with the results of the proteome, showing a trend of first rising and then falling (Figure 5H). Generally, these tunnel-based flow cytometry and qRT-PCR results agreed with those analyzed by the proteome.


*2.5. qRT-PCR Validation*


Based on the expression results of proteomic data, eight significantly upregulated or downregulated genes were selected for qRT-PCR validation, including *Fibrinogen-like protein A*, *Active RNA polymerase*, *HEBP2*, *Exosome component 10*, *FALDH*, *TS*, *CRIM1*, *ECI1* (Figure 6). The results showed that except for *ECI1*, the transcripts of the other seven genes showed a regulatory trend consistent with the proteome during intestinal regeneration.

## 3. Discussion

This study used TMT-based quantitative proteomics to construct protein-expression profiles associated with various stages of intestinal regeneration in *A. japonicus*. Although proteomic analysis has been conducted on the normal intestine and the early-regenerated stage intestine (3-dpe), an increasing number of studies have shown that the swelling of the free end of the mesentery was necessary for the formation of the new intestine and served as the epicenter of intestinal regeneration [15,16]. Therefore, in order to comprehensively and systematically explore the mechanism of *A. japonicus* intestinal regeneration, we treated the mesentery and regenerated intestine as a whole and detected DEPs and regulatory pathways throughout the entire stage of intestinal regeneration.

We performed enrichment analyses of differential proteins in normal and regenerative stages based on KEGG annotation to investigate the functional distribution of identified proteins. The results revealed that pathways involved in multiple amino acids and metabolic cascade reactions were upregulated during the process of intestinal regeneration, such as glutathione metabolism, cysteine, and methionine metabolism. Focusing on glutathione metabolism, a central component in the redox balance, we found that it was highly enriched in the early stage (2 dpe) of intestinal regeneration and speculate that the organism upregulates glutathione metabolism to balance the free radicals produced by tissue damage and wound repair, enabling cells to achieve oxidative balance. Similarly, the upregulation of glutathione metabolism has also been confirmed in other regeneration modes. We found that it is highly present in planarians and that a significant reduction in glutathione content led to regenerative failure with tissue lesions [19]. Likewise, increased glutathione metabolism level occurs early during liver regeneration, preventing the increase in GSH after a two-third partial hepatectomy will blunted liver regeneration [20]. Focusing on cysteine and methionine metabolism, a mechanism that interacts with other amino acids such as glycine, alanine, and leucine through a series of enzymatic reactions, playing an important role in maintaining amino acid balance, protein synthesis, and many biochemical reactions in the body. In this research, cysteine and methionine metabolism were significantly upregulated and enriched during the intestinal regeneration stage from 7-dpe to 28-dpe, indicating that they play an important role in regulating the intestinal regeneration process in the middle and late stages. It was suggested the regulation of cysteine and methionine metabolism during these periods contributes to the degradation and synthesis of proteins, facilitates the transformation of various fate states in newly formed cells, and enables smooth regeneration.

In order to better investigate the expression trends of differential proteins throughout the intestinal regeneration process and the cytological regulatory processes involved, the differential proteins were clustered according to different expression trends and then analyzed by GO enrichment analysis in this study. The interesting point in the results is that whether the proteins in cluster 1 and cluster 2 showed a gradual up-regulation during intestinal regeneration, the proteins of cluster 3 showed a gradual down-regulation or the proteins of cluster 4 showed a down-regulation followed by an up-regulation, or the proteins of cluster 5 and cluster 6 showed a trend of up-regulation followed by a down-regulation, the proteins involved in the various metabolic processes were significantly enriched. As is well known, metabolic characteristics are not static during growth and regeneration but rapidly switch based on cellular needs [21,22]. Therefore, we speculate that the upregulation or downregulation trend of a large number of metabolic pathways in Figure 4 was related to the complex cell fate changes involved in the intestinal regeneration process. It is worth noting that these metabolic pathways serve the key purpose of transforming or utilizing energy to maintain cell integrity and survival and play a crucial role in recombinant gene expression to determine cell identity and function during regeneration by affecting cell signaling and epigenetic regulators [23,24].

Previous proteomic research on the intestine regeneration of *A. japonicus* primarily concentrated on the differential expression proteins between the early regeneration phase (3-dpe) and the normal state [16]. This research has revealed a wealth of critical signaling pathways and differentially expressed proteins involved in the early regulatory processes of intestinal regeneration, such as cytoskeletal proteins, energy metabolism, and the interplay between ECM and its receptors, which further offer substantial insights into comprehending the initial mechanisms of intestine regeneration. However, *A. japonicus* intestine regeneration is a complex multistage process that can be divided into four stages: the initial phase of wound healing (days 0–2), the stage of blastema formation (days 2–12), the stage of lumen formation (days 12–28), and subsequent growth phases [15]. Each stage involves specific cellular events and ordered regulatory processes that play a crucial role in the final formation of the intestine structure. Therefore, studying the signal regulatory mechanisms at each critical stage of intestine regeneration is of great significance for fully revealing the regulation mechanism of intestine regeneration. In addition, preliminary studies have indicated that sea cucumber intestine development always occurs in the expanded regions of the mesentery’s free edge, considered the core area for intestine repair [6,25]. However, previous research on *A. japonicus* focused primarily on the location where regenerative intestine cavities form; relatively little attention has been paid to the mesentery and its unpenetrated free end [26,27]. Therefore, in this experiment, we treated the mesentery and the regenerative intestinal primordium as a whole for analysis and conducted TMT proteomics sequencing of the key periods during the intestine regeneration process of *A. japonicus*. This approach allows us to obtain differentially expressed proteins on the mesentery and its free endogenous regenerative primordium and combines the differential proteins at various stages of gut regeneration with specific tissue morphological changes and cellular fate changes, enabling us to screen for key regulatory pathways.

Cell proliferation and cell apoptosis are two key processes that control the number of cells in multicellular organisms [28]. Although the balance between cell proliferation and cell apoptosis is strictly controlled, during certain special physiological processes such as wound repair, tissue development, and tissue regeneration, the body breaks this balance, changing the number of cells to meet the needs of the organism [29]. Current research on cell proliferation and cell apoptosis during sea cucumber intestinal regeneration is still at an early stage. In the present study, our results showed that cell proliferation and apoptosis-related proteins were significantly upregulated in the middle and late stages of intestinal regeneration by analyzing DEPs throughout the intestinal regeneration process. Subsequently, flow cytometry analysis using EdU- and Tunel- techniques revealed that changes in cell proliferation and apoptosis phenotypes were consistent with the trends observed in protein expression. In our previous study, morphological data showed that primary intestinal cells undergo quite dense cell division and demonstrated that the entry and progression of the cell cycle are particularly important for regulating intestinal regeneration [15]. At the same time, a large amount of cell apoptosis was also found in the intestinal regeneration of sea cucumbers, proving that cell apoptosis is an important process in regulating intestinal regeneration. In addition, similar results were also detected in other species of sea cucumber, like *Holothuria glaberrima* (*H. glaberrima*) [13,30]. Mashanov et al. employed the 5-bromo-2’-deoxyuridine (BrdU) and Tunel techniques to monitor cellular proliferation and apoptosis during the regenerative process of the intestine in *H. glaberrima* [30]. The research revealed that in normal conditions, cellular proliferation and apoptosis rates in the tissues are rather minimal, functioning merely to preserve regular cellular functions and the cell cycle’s continuity. However, during the process of intestinal regeneration, a large amount of cell proliferation and apoptosis were detected on the free end of the mesentery of *H. glaberrima*, and their quantity changes were positively correlated with regulation, and they were most active in the stages of intestinal lumen formation and intestinal differentiation [31]. These data reveal that different types of sea cucumbers may share some common trends in cell proliferation and apoptosis during intestinal regeneration. Specifically, these trends are not apparent at the early stages of intestinal regeneration but become very significant from the middle to late stages of regeneration. Although it is generally believed that the balance between cell proliferation and apoptosis during regeneration tends to favor cell proliferation, apoptosis also plays an equally important role in tissue regeneration.

The interaction between cell proliferation and apoptosis has made significant contributions to various physiological processes that occur during tissue remodeling, regeneration, and morphogenesis. Multiple studies indicate that apoptosis acts as a signal to stimulate promotion within the regenerative issues, producing the cells needed for full regeneration [32]. The conservation of methodology as a regenerative mechanism demonstrated specific highlights of its importance and motivation for the ongoing investment of this important facet of programmed cell death. Josean et al. investigated the contributions of cell proliferation and apoptosis during the early stages of sea cucumber intestinal regeneration using pharmacologically defined regulators of cell death (zVAD) and cell proliferation (aphidicolin) [17]. They then evaluated the process of intestinal regeneration by measuring the thickness of the mesenterial free ends. The results indicated that partially inhibiting apoptosis significantly affected cell proliferation during the formation of the gut primordium, while partially inhibiting cell proliferation did not affect cell apoptosis during the formation of the gut primordium. This phenomenon is similar to a process called apoptosis-induced proliferation (AiP) in model organisms [33]. In AiP, cells undergoing apoptosis release and secrete signaling factors promoting neighboring cell proliferation [34,35,36]. For example, during hydra head regeneration, apoptotic cells release mitogenic signal Wnt3, triggering adjacent cell proliferation [37]. Inhibiting apoptosis can prevent the release of Wnt ligands, thereby inhibiting active cell division and interrupting head regeneration processes. Similarly, in clawed frog tadpole tail regeneration, inhibiting apoptotic mediator caspase-3 inhibited cell proliferation, thus blocking tail regeneration [38]. Similar situations also occur in mouse wound healing and liver regeneration processes, where cysteine protease inactivation leads to the loss of liver regeneration capacity [39,40]. This study shows preliminary evidence of an AiP phenotype similar to that found in higher organism regeneration models during the intestine regeneration processes of *A. japonicus* and other types. However, further research is needed to fully validate these phenotypes and explore their potential regulatory mechanisms. For future development, we hope to apply the findings on the interactions between cell proliferation and cell death in lower organisms to human tissue regeneration research, such as studies on liver, intestine, and skin regeneration, thereby promoting the development of human regenerative medicine.

## 4. Materials and Methods

### 4.1. Animals and Samples

Healthy *A. japonicus* (80–110 g) were collected from a farm in Dalian, Liaoning province, China, and acclimated in seawater aquaria at 11 °C. After acclimation, *A. japonicus* used for intestinal regeneration experiments were eviscerated by injecting 3–5 mL 0.35 M KCl into the coelom [13,15], and un-eviscerated animals (controls) were kept under the same conditions and fed once daily. The regeneration tissues, including mesentery and/or intestinal rudiment from 9 individuals, were sampled from 2, 7, 12, 20, 28 dpe group, respectively. The mesentery of healthy *A. japonicus* without any treatment was also sampled and served as the control. The dissected normal tissue and regenerated primordia were frozen and stored in liquid nitrogen.

### 4.2. Protein Preparation

To extract total protein, tissue samples weighing 100 mg were obtained from each normal and various regenerating groups (2, 7, 12, 20, 28 dpe), ground into powder after being flash-frozen in liquid nitrogen. This powder was then completely solubilized in 1 mL of Radio Immunoprecipitation Assay (RIPA) lysis buffer (50 mM Tris (pH 7.4), 150 mM NaCl, 1% TritonX-100, 1% sodium deoxycholate, 0.1% SDS, 2 mM sodium pyrophosphate, 25 mM β-glycerophosphate, 1 mM EDTA and 1 mM Na_3_VO_4_) (Beyotime, Shanghai, China) and mixed it thoroughly with phenylmethylsulfonyl fluoride (PMSF) (Merck, Darmstadt, Germany) at a final concentration of 1 mM. Following a 5 min interval, an additional 10 mM Dithiothreitol (DTT) (final concentrations) was added to the samples. The mixture was sonicated for 5 min (15 kHz, 10 s pulses, and 10 s rest) before being centrifuged at 4 °C and 12,000× *g* for 15 min. The resulting supernatant was incubated at 95 °C for 8–15 min, then placed on ice for 2 min, and centrifuged at 4 °C, 12,000× *g* for 15 min. To break down the disulfide bridges within the proteins present in this supernatant, 10 mM DTT was added, and the solution was incubated at 56 °C for 1 h. Subsequently, 55 mM Iodoacetamide (IAM) (Biosharp, Guangzhou, China) was added to the sample to cap cysteine residues, and the mixture was left to stand in darkness for at least 2 h. The supernatant was mixed with 55 volumes of cold acetone and refrigerated at −20 °C for 2 h to induce protein precipitation. After centrifuging at 4 °C and 12,000× *g* for 15 min, the supernatant was discarded, and the pellet was air-dried for 5 min. It was then re-dissolved in 500 μL of Dissolved Buffer (DB buffer) (8 M Ureaurea, 100 mM TEAB, pH = 8.5). Samples were centrifuged once more at 4 °C and 12,000× *g* for 15 min. The clear supernatant was moved to a fresh tube, and its quantity was determined through quality assessment procedures. The proteins contained within this supernatant were preserved at −80 °C for further scientific evaluation.

The protein quantification of each group sample was performed using the Bradford protein assay kit (Thermo Scientific, Waltham, MA, USA) according to the manufacturer’s instructions. First, a standard solution of bovine serum albumin (BSA) proteins with concentration gradients ranging from 0–0.5 µg/µL was prepared. Then, different concentrations of BSA standard protein solutions and various dilution factors of sample solutions were added to a 96-well plate, filling the volume to 20 µL each, repeating the gradient three times. Next, quickly add 180 µL of G250 dye solution, place at room temperature for 5 min, and measure the absorbance at 595 nm. Draw a standard curve using the standard protein solution’s absorbance and calculate the protein concentration of the test samples. To assess the quality of proteins, 20 µg protein from each group was taken for sodium dodecyl sulfate-polyacrylamide gel electrophoresis (SDS-PAGE) (12%) analysis. This included concentrating gel electrophoresis at 80 V for 20 min and separating gel electrophoresis at 120 V for 90 min. Following electrophoresis, Coomassie Brilliant Blue R-250 (Beyotime, Shanghai, China) staining was performed, and the bands were cleared after bleaching.

Enzymatic hydrolysis and desalination are required for the protein. Firstly, DB buffer solution was added to each group of protein extraction solutions to reach a total volume of 100 μL separately. Then, Trypsin (Solarbio, Beijing, China) and a 100 mM Tetraethylammonium Bicarbonate (TEAB) buffer solution (Merck, Darmstadt, Germany) were added, mixed thoroughly, and digested at 37 °C for 4 h. Next, additional trypsin and CaCl_2_ were added for overnight digestion. Following that, the pH was adjusted to below 3 with formic acid, mixed, and centrifuged at room temperature at 12,000× *g* for 5 min. The supernatant was collected slowly through a C18 desalting column, washed three times with washing solution (0.1% formic acid, 3% acetonitrile), and finally, an appropriate amount of elution solution (0.1% formic acid, 70% acetonitrile), collected the filtrate and freeze dry it.

### 4.3. TMT Labeling

100 μg protein separately obtained from normal and different regenerating groups (2, 7, 12, 20, 28 dpe) were subjected to digestion using Trypsin Gold (Promega, Madison, WI, USA) at a ratio of protein to trypsin of 30:1 at 37 °C for 16 h. Following trypsin digestion, the resulting peptides were dehydrated through vacuum centrifugation, reconstituted in 0.5 M TEAB, and further processed as per the manufacturer’s protocol for the 8-plex TMT reagent (Applied Biosystems, Waltham, MA, USA). A single unit of TMT reagent was defrosted and dissolved in 24 μL of isopropanol. The peptides were tagged with isobaric labels and left to stand at room temperature for 2 h. The tagged peptide mixtures were combined and dehydrated once again via vacuum centrifugation.

High pH reversed-phase fractionation TMT labeled peptides from 6 groups was desalted with a C18 column (Dr. Maisch, Ammerbuch, Germany). Following that, they were gathered and vacuum-dried to prevent elution below 4 °C. For RP-RP fractionation, peptides were again suspended in buffer A (2% acetonitrile in 0.1% formic acid). Using Thermo Scientific, USA’s High pH Reverse Phase Fractionation (hpRP) chromatography, 100 μg of peptides were fractionated at a pH of 10. The following parameters were set: 150 × 2.1 mm chromatographic column (Waters, Milford, MA, USA, XBridge BEH C 18 XP Column); 10 mM Formic Acid (FA) at pH = 10 in mobile phases A and B; 10 mM FA, 90% Acetonitrile (ACN), 10% H_2_O, pH = 10 in mobile phases B. Gradients were used for 120 min to separate the samples. 5–8% for 5 min, 8–18% for 35 min, 18–32% for 22 min, 32–95% for 2 min, 95% for 4 min, and 95–100% for 4 min. Peptides were fractionated to 180 fractions (40 s intervals) and then were combined to 20 fractions. Samples were vacuum-dried and stored at −80 °C before LC-MS/MS analysis.

### 4.4. LC-MS/MS Analysis Based on Orbitrap Exploris 480

Using the Orbitrap Exploris 480 mass spectrometer (Thermo Scientific, Waltham, MA, USA) with a Nanospray Flex™ (ESI) ion source, set the ion spray voltage to 2.1 kV and the ion transfer tube temperature to 320 °C. The mass spectrometry data acquisition mode is based on data dependency, with FAIMS compensation voltages set to −45 V and −65 V. Both voltages use identical acquisition parameters: full MS scanning range from *m*/*z* 350–1500, primary MS resolution set at 60,000 (200 *m*/*z*), C-trap maximum capacity set to Auto (software automatically calculates optimal capacity based on other settings), C-trap maximum injection time set to Auto, maintaining highest sensitivity while keeping maximum scan speed for automatic calculation of ion injection time; secondary loop scan time is 1 s, using high-energy collision dissociation (HCD) method within 1 s to fragment as many precursor ions as possible in descending order of response intensity from the first MS scan, secondary MS resolution set at 30,000 (200 *m*/*z*) turning on turbo TMT + preprocessing function, C-trap maximum capacity set to 1 × 10^5^, C-trap maximum injection time set to Auto, peptide fragmentation collision energy set to 36%, threshold intensity set to 5.0 × 10^3^, dynamic exclusion range set to 45 s, generating MS raw data files (.raw).

### 4.5. Identification and Quantification of Proteins

The search software Proteome Discoverer version 2.4 (PD, Thermo, HFX, and 480) was applied to search for the resulting spectrum of each run based on the *A. japonicus* transcriptomes [41,42]. This process involves searching each run’s result spectrum with PD, setting specific parameters for precursor ion mass tolerance at 10 ppm and fragment ion mass tolerance at 0.02 Da. Fixed modifications include cysteine alkylation, while variable modifications include oxidation of methionine in peptide side chains, TMT labeling of peptide side chains, TMT labeling of N-termini, acetylation, loss of methionine, and loss of methionine plus acetylation. Up to two missed cleavage sites are allowed.

To improve the quality of analytical results, PD software further filters the retrieval results: Peptide Spectrum Matches (PSMs) with credibility above 99% are considered credible PSMs, while proteins containing at least one unique peptide segment are considered credible proteins. Only credible PSMs and proteins are retained, and FDR validation is performed to remove peptide and protein segments with an FDR greater than 1%. *t*-test analysis is conducted on the quantitative results of proteins, defining proteins with significant quantitative differences between experimental and control groups (*p* < 0.05, |log2FC| > * (FC > * or FC < * [fold change, FC]) as differentially expressed proteins (DEPs).

### 4.6. Cluster Analysis Based on Protein Functional Enrichment

Cluster analysis based on functional enrichment was utilized to explore potential connections and differences in specific functions, including GO and KEGG pathways. After collecting functional categorization information and *p*-values, functional classes with substantial enrichment (*p* < 0.05) in at least one protein cluster were selected. The filtered *p*-value matrix underwent a logarithmic transformation of −log10, followed by a Z transformation for each functional categorization. Finally, the Z-transformed dataset was examined using hierarchical clustering (Euclidean distance and average linked clustering). The logical relationship of DEPs among normal and various regeneration groups was analyzed by Venn diagram (Venn plot in the R package (v3.6.0)). The expression trends of DEPs among normal and various regeneration groups were enrichmented analysis using clusterProfiler in the R package (v3.6.0) package, followed by plotting using enrichplot and ggplot2.

### 4.7. Heat Map Analysis

The clustering relationship was displayed using heatmap.2 in the R package (v3.6.0). Firstly, it is necessary to establish a database of differentially expressed proteins at various time points during the intestine regeneration process of *A. japonicus* (2, 7, 12, 20 and 28 dpe). Then, combine with KEGG annotation to screen out DEPs related to the different signaling pathways. After further removing undefined hypothetical proteins, the expression trends of proteins from each signaling pathway were plotted using the R package “heatmap.2” and “ggplot2” during the regeneration process [43].

### 4.8. Quantitative Real-Time PCR Analysis

According to the manufacturer’s instructions, total RNA was extracted from tissues (50 mg) of normal and different regeneration groups (2, 7, 12, 20, and 28 dpe) using 1 mL Trizol (Sigma, Saint Louis, MO, USA) with 3 parallel sets for each group. The concentration of the RNA was detected by NanoDrop 8000 (Thermo Scientific, Waltham, MA, USA), and the integrity of the RNA was evaluated by 1.5% agarose gel. To ensure complete removal of genomic DNA (gDNA), 1 μg of total RNA was incubated with 1 unit of DNase I (Sigma, Saint Louis, MO, USA) for 15 min at room temperature. Complementary DNA (cDNA) was synthesized using a Reverse Transcriptase M-MLV kit (TaKaRa, Shiga, Japan) according to the manufacturer’s instructions and stored at −20 °C for RT-qPCR.

The specific primers for genes were designed by Primer Premier 5 software according to its CDS with the following parameters: melting temperatures of 56–62 °C, primer lengths of 18–22 bp, and product lengths of 150–250 bp [44]. The primer details are listed in Table 2. The expression profiles of genes were detected by SYBR Green real-time PCR assay. Specifically, the cDNA concentrations of each group were diluted 50-fold as a template for RT-qPCR. Amplification was conducted in a 20 μL reaction volume containing 10 μL of SYBR Green I Master (TaKaRa, Shiga, Japan), 0.4 μL each of forward and reverse primers, 2 μL of diluted cDNA and 7.2 μL of RNase-free water. The thermal cycling profile consisted of an initial denaturation at 95 °C for 30 s, followed by 45 cycles of denaturation at 95 °C for 5 s and extension at 60 °C for 30 s. Each assay was performed in triplicate with the β-tubulin gene as the endogenous control. All data were analyzed relative to the β-tubulin gene by the 2^−ΔΔCt^ method (mean ± SD, *n* = 5). Analysis of variance followed by multiple Duncan tests was conducted to discern differences in mean values between the control and experimental groups. Differences among normal and different regeneration groups were marked with an asterisk for * *p* < 0.05, ** *p* < 0.01, *** *p* < 0.001, **** *p* < 0.0001. Error bars within the graphs signified standard deviations.

### 4.9. Flow Cytometry Assay

Ethynyl-20-deoxyuridine (EdU)-based and Tunel-based flow cytometry methods were performed to the quantitative dynamic trend of cell proliferation and apoptosis throughout the intestinal regeneration. For cell proliferation assay, 2 mM EdU (Beyotime, Shanghai, China) was administered through an intraperitoneal injection 24 h prior to each collection time point during the process of intestinal regeneration. Cell suspensions from each tissue sample at various stages of regeneration were prepared by enzymatic digestion with Collagenase Type IV and Trypsin (Gibco, Grand Island, NY, USA). These cellular preparations were subjected to centrifugation at 800× *g* for 5 min and then resuspended in 4% Paraformaldehyde (Solarbio, Beijing, China) for 15 min. Following three rinses with PBST (consists of 2.2 M NaH_2_PO_4_, 8.1 M Na_2_HPO_4_, 137 mM Nacl, 2.7 mM KCl, and 0.05% Tween 20), the cells were treated with EdU Click Reaction Solution (Beyotime, Shanghai, China) for 30 min at 37 °C in the dark. After the final wash, the proliferation index of these cells (consisting of 10^5^ cells) was determined using a flow cytometer (Miltenyi Biotech, Bergisch Gladbach, Germany). Data analysis was performed using Flow Jo software version 10.6.1 (BD Biosciences, Franklin Lakes, NJ, USA). For apoptosis assay, we prepared and obtained single-cell suspensions from each regenerated group according to the above-mentioned dissociation method, then placed the obtained single-cell suspension cells in 4% paraformaldehyde for fixation for 15 min, followed by permeabilization with 0.3% (*v*/*v*) triton X-100 (Merck, Darmstadt, Germany) for 10 min. After washing with PBST three times, these cells were incubated with TUNEL detection reagent (Beyotime, Shanghai, China) and incubated at 37 °C in the dark for 60 min. After the last wash in a triple, the percentage of apoptosis rate of these 1 × 10^5^ cells was measured by a flow cytometer (Miltenyi Biotech, Bergisch Gladbach, Germany) and analyzed by FlowJo software v10.6.1 (BD Biosciences, Franklin Lakes, NJ, USA).

### 4.10. Data Analysis

All statistical analyses were conducted using the GraphPad Prism program (GraphPad Software, Version 9). A one-way analysis of variance (ANOVA) was employed to identify significant differences between the control and experimental groups regarding the percentage of EDU-positive and Tunel-positive cell populations. These findings were illustrative of at least three independent experiments and were presented as mean ± (SD). The significance levels were defined as follows: * *p* < 0.05, ** *p* < 0.01, *** *p* < 0.001, **** *p* < 0.0001.

## 5. Conclusions

In this study, we used TMT-based quantitative proteomics to construct protein-expression profiles associated with various stages of intestinal regeneration in *A. japonicus*. Our findings revealed that approximately 538, 445, 397, 1012, and 966 DEPs were detected (*p* < 0.05) between the normal and 2-, 7-, 12-, 20, and 28 dpe stages, respectively, involving many signal pathways, such as cell proliferation and apoptosis. Further EdU- or Tunel-based flow cytometry assay and western blotting experiments not only confirmed that during intestinal regeneration, the expression levels of proteins related to cell proliferation showed an increasing trend but also revealed that the expression levels of proteins associated with cell apoptosis experienced a pattern of initially increasing followed by decreasing. These dynamic changes in cellular phenotypes are consistent with the expression trends of corresponding proteins, which provide an important basis for studying the regulatory mechanisms associated with intestinal regeneration in sea cucumbers.

## Figures and Tables

**Figure 1 ijms-25-04250-f001:**
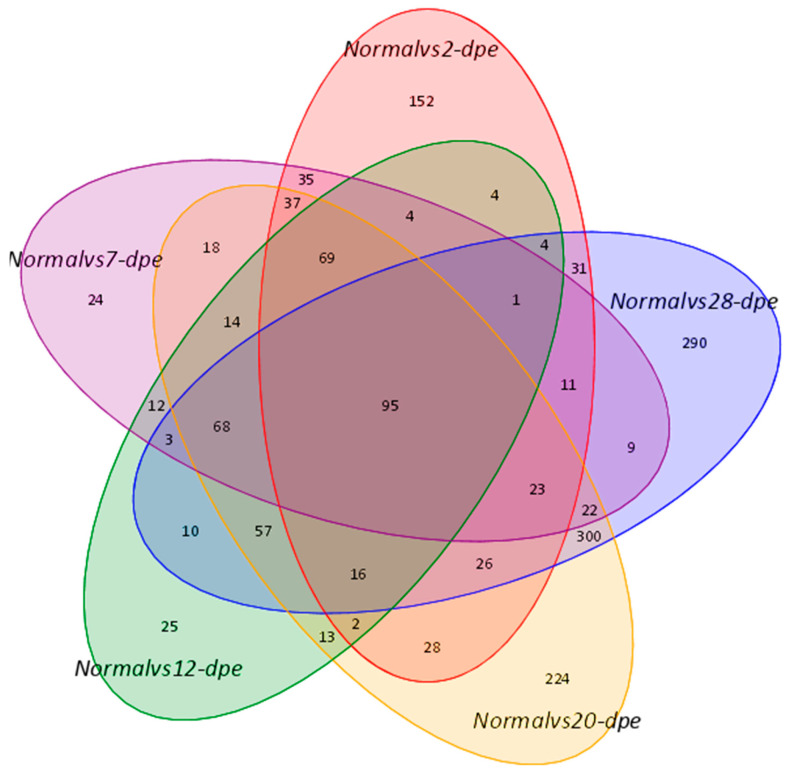
A Venn diagram showed overlapping genes between the normal group and different regenerative groups.

**Figure 2 ijms-25-04250-f002:**
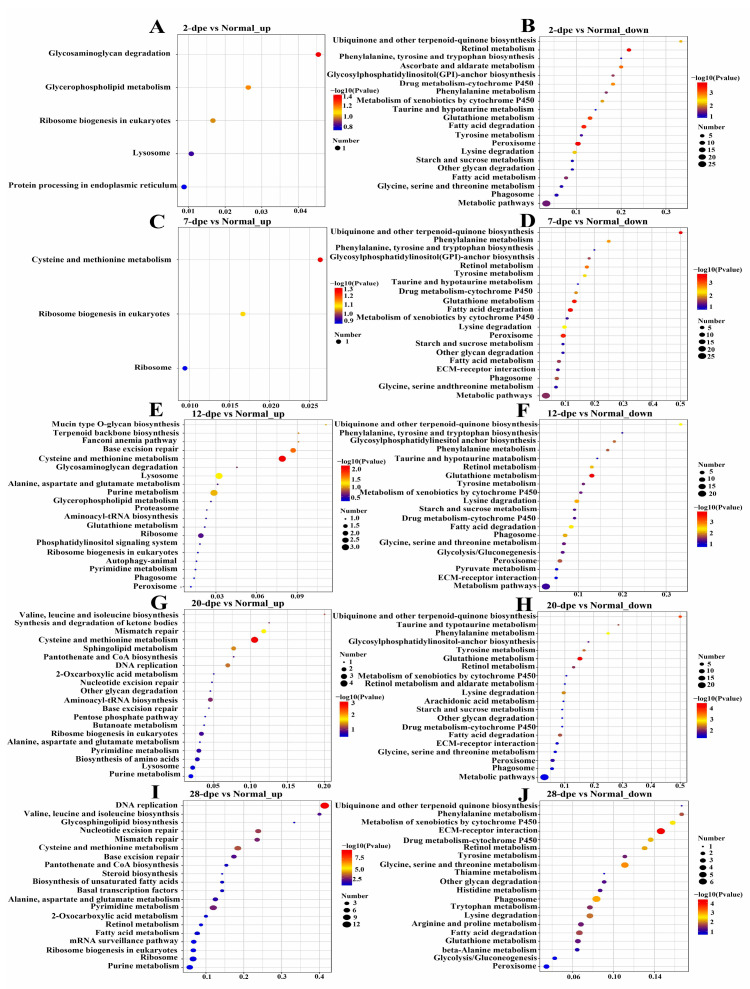
Kyoto Encyclopedia of Genes and Genomes pathway enrichment analysis of differentially expressed proteins in normal and different regeneration groups. (**A**) 2 dpe vs. Normal up-regulation. (**B**) 2 dpe vs. Normal down-regulation. (**C**) 7 dpe vs. Normal up-regulation. (**D**) 7 dpe vs. Normal down-regulation. (**E**) 12 dpe vs. Normal up-regulation. (**F**) 12 dpe vs. Normal down-regulation. (**G**) 20 dpe vs. Normal up-regulation. (**H**) 20 dpe vs. Normal down-regulation. (**I**) 28 dpe vs. Normal up-regulation. (**J**) 28 dpe vs. Normal down-regulation.

**Figure 3 ijms-25-04250-f003:**
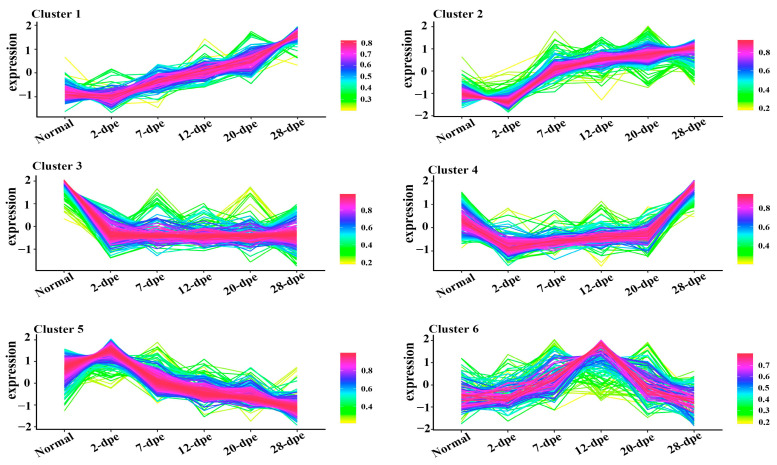
Trend analysis of differentially expressed proteins into six clusters. The lines represent the expression trend of each protein in the normal and five regeneration groups (2 dpe, 7 dpe, 12 dpe, 20 dpe, 28 dpe).

**Figure 4 ijms-25-04250-f004:**
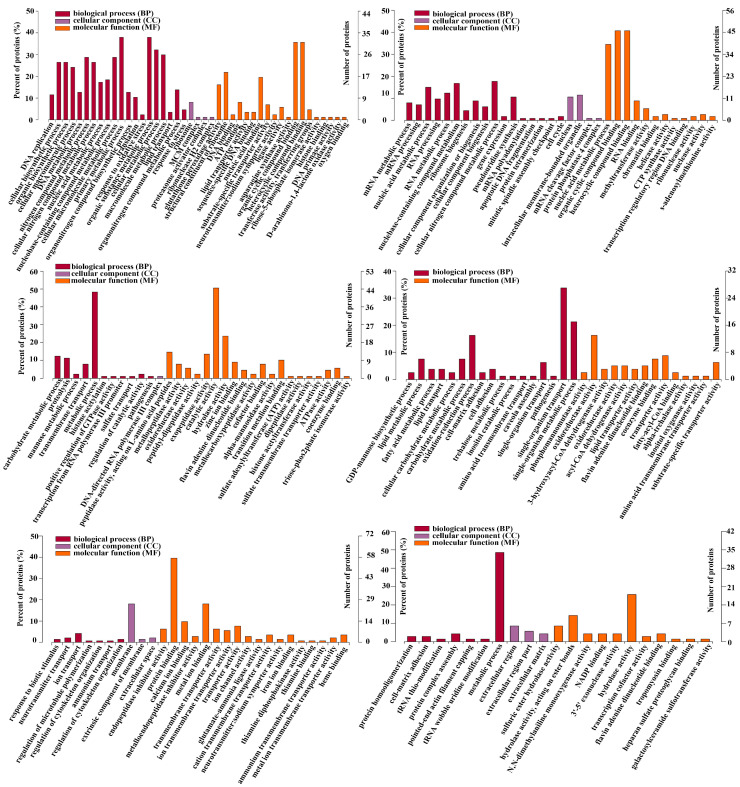
Gene ontology enrichment analysis of DEPs of six clusters (Figure 3). Ontology domains comprise biological process (BP), cellular component (CC), and molecular function (MF).

**Figure 5 ijms-25-04250-f005:**
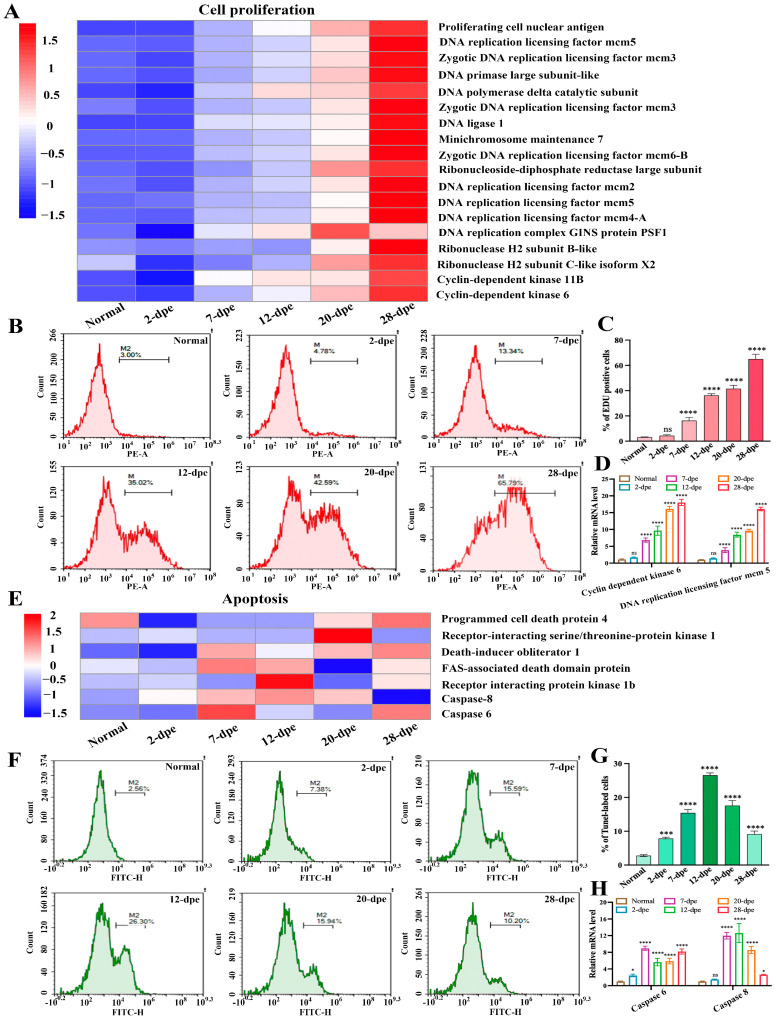
The expression patterns of proteins related to different cell fate states during the process of intestine regeneration. (**A**) The expression heatmap of cell proliferation-related proteins during the process of intestinal regeneration. (**B**) EdU-based flow cytometry was used to analyze the changes in cell proliferation levels of normal and different regenerative stages (2–28 dpe). (**C**) The number of EdU-labeled cells corresponding to (**B**). Results are the mean ± SD from three independent experiments. “ns” indicates no significant difference versus the control. **** *p* < 0.0001 versus the control. (**D**) qRT-PCR analysis of the mRNA expression of cell proliferation-related genes (Cyclin-dependent kinase 6 and DNA replication licensing factor mcm 5). The data are presented as mean ± SD, *n* = 3 repeats. “ns” indicates no significant differences (*p* > 0.05), **** *p* < 0.0001. (**E**) The expression heatmap of apoptosis-related proteins during the process of intestinal regeneration. (**F**) Flow cytometric detection of apoptosis in the process of intestine regeneration using the tunel technique. (**G**) The number of tunnel-labeled cells corresponding to (**F**). Results are the mean ± SD from three independent experiments. *** *p* < 0.001, **** *p* < 0.0001 versus the control. (**H**) qRT-PCR analysis of the mRNA expression of apoptosis-related genes (Caspase 6 and Caspase 8). The data are presented as mean ± SD, *n* = 3 repeats. “ns” indicates no significant differences (*p* > 0.05), * *p* < 0.05, *** *p* < 0.001, **** *p* < 0.0001.

**Figure 6 ijms-25-04250-f006:**
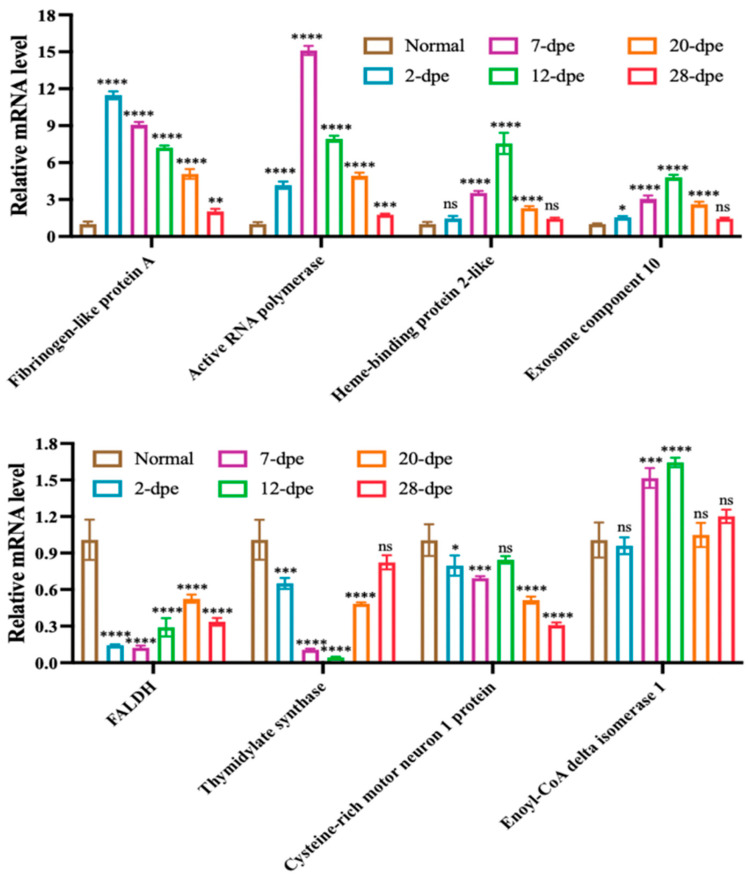
Quantitative RT-PCR analysis of differentially expressed genes of *A. japonicus* during the process of intestine regeneration. The data are presented as mean ± SD, *n* = 3 repeats. “ns” indicates no significant differences (*p* > 0.05), * *p* < 0.05, ** *p* < 0.01, *** *p* < 0.001, **** *p* < 0.0001.

**Table 1 ijms-25-04250-t001:** Differential expression of proteins of *A. japonicus* in different stages of intestinal regeneration.

Group	Protein Description	Fold Change		*p*-Value
		Down	UP	
Normal VS 2-dpe	Fibrinogen-like protein A (FGLA)		17.93	1.99 × 10^−105^
	Subtilisin-like protease 1 (SUB1)		6.12	3.80 × 10^−43^
	Solute Carrier Family 6 Member 14 (SLC6A14)		5.72	3.13 × 10^−40^
	Ficolin-2 (FCN-2)		2.80	3.53 × 10^−21^
	Tripartite motif-containing 71 (TRIM71)		2.76	3.96 × 10^−15^
	Fatty aldehyde dehydrogenase (FALDH)	0.02		5.25 × 10^−109^
	Angiotensin-converting enzyme (ACE)	0.07		7.68 × 10^−54^
	Thymidylate synthase (TS)	0.07		6.59 × 10^−53^
	Fatty acid-binding protein 2 (FABPI)	0.09		5.21 × 10^−43^
	Enoyl-CoA delta isomerase 1 (ECI1)	0.09		5.80 × 10^−43^
Normal VS 7-dpe	Subtilisin-like protease 1 (SUB1)		25.08	1.26 × 10^−70^
	Caspase-6 (CASP6)		7.63	9.27 × 10^−29^
	Fibrinogen-like protein A (FGLA)		5.38	4.04 × 10^−20^
	Heme-binding protein 2 (HEBP2)		4.67	4.70 × 10^−17^
	Uronyl 2-sulfotransferase (UST)		4.19	6.56 × 10^−15^
	Fatty aldehyde dehydrogenase (FALDH)	0.03		5.07 × 10^−70^
	Enoyl-CoA delta isomerase 1 (ECI1)	0.09		5.45 × 10^−33^
	Thymidylate synthase (TS)	0.09		2.15 × 10^−31^
	Fatty acid-binding protein 2 (FABPI)	0.10		6.42 × 10^−31^
	Long-chain specific acyl-CoA dehydrogenase (ACADL)	0.11		7.97 × 10^−27^
Normal VS 12-dpe	Heme-binding protein 2 (HEBP2)		6.51	6.71 × 10^−16^
	Mucin-2 (MUC2)		4.76	1.97 × 10^−11^
	Transmembrane prolyl 4-hydroxylase (P4HTM)		4.43	1.58 × 10^−10^
	Exosome component 10 (EXOSC10)		4.34	2.87 × 10^−10^
	Dimethylaniline monooxygenase (FMO)		4.08	1.53 × 10^−09^
	Fatty aldehyde dehydrogenase (FALDH)	0.03		8.82 × 10^−47^
	Angiotensin-converting enzyme (ACE)	0.08		6.13 × 10^−24^
	Enoyl-CoA delta isomerase 1 (ECI1)	0.09		3.53 × 10^−22^
	Fatty acid-binding protein 2 (FABPI)	0.10		9.63 × 10^−22^
	Thymidylate synthase (TS)	0.11		4.14 × 10^−19^
Normal VS 20-dpe	Subtilisin-like protease 1 (SUB1)		15.96	9.67 × 10^−28^
	Fibrinogen-like protein A (FGLA)		6.25	7.18 × 10^−13^
	sulfotransferase family cytosolic 1B member 1 (SULT1B1)		4.15	2.66 × 10^−08^
	Uronyl 2-sulfotransferase (UST)		3.89	1.14 × 10^−07^
	Heme-binding protein 2 (HEBP2)		3.47	1.17 × 10^−06^
	Fatty aldehyde dehydrogenase (FALDH)	0.03		2.19 × 10^−39^
	Angiotensin-converting enzyme (ACE)	0.08		7.64 × 10^−22^
	Enoyl-CoA delta isomerase 1 (ECI1)	0.09		8.07 × 10^−20^
	Fatty acid-binding protein 2 (FABPI)	0.10		4.11 × 10^−18^
	Thymidylate synthase (TS)	0.13		8.75 × 10^−15^
Normal VS 28-dpe	Elongation of very long chain fatty acids protein 6 (ELOV6)		8.13	1.76 × 10^−10^
	Peroxisomal carnitine O-octanoyltransferase (CROT)		6.61	8.66 × 10^−09^
	Tryptophan–tRNA ligase (TrpS)		6.59	9.06 × 10^−09^
	Transducin beta-like protein 3 (TBL3)		5.38	2.87 × 10^−07^
	Acyl-CoA-binding protein (ACBP)		5.17	5.29 × 10^−07^
	Enoyl-CoA delta isomerase 1 (ECI1)	0.08		1.63 × 10^−15^
	Angiotensin-converting enzyme (ACE)	0.10		1.22 × 10^−12^
	Fatty aldehyde dehydrogenase (FALDH)	0.12		1.69 × 10^−11^
	Cysteine-rich motor neuron 1 (CRIM1)	0.14		4.12 × 10^−10^
	Fatty acid-binding protein 2 (FABPI)	0.14		5.73 × 10^−10^

**Table 2 ijms-25-04250-t002:** qRT-PCR primers used in this study.

Primer	Sequence (5′-3′)
*β-Tubulin*	F: GCACATCAAGCCGTCAAACTCAC
	R: TATGCCCGCATAGCAAACATACC
*Fibrinogen-like protein A*	F: CAGTCAGGTCTTTCGTGTCCC
	R: GTCCATCCTCCCTCGTCAGTT
*Active RNA polymerase*	F: TGTTGATAGTGGTGATTCC
	R:CCCCTTTTTGCCTGGCTTC
*Heme-binding protein 2-like*	F: CCTGTTTCTGGAGTATTCG
	R: CTCTGTTCAGTTTGTTGGC
*Exosome component 10*	F: TCAGCAGAGAATGGAGTTT
	R: CGAGGGTATCAATGAGGTA
*FALDH*	F: GAGATTTTCGGTCCTCTAT
	R: AGGCACTGTTGCTTGTATT
*Thymidylate synthase*	F: AAAGTGTTTTCTGGAGGG
	R: TGGGTTGGTTTTAATCGTC
*Cysteine-rich motor neuron 1 protein*	F: CATCAGCTACTGCACCATA
	R: CTAGACAGCATTCACCCTC
*Enoyl-CoA delta isomerase 1*	F: AAAAGTGGGTATTGTGGCT
	R: CCTGGTGGAGTTTGGATTG
*Caspase-6*	F: TACCTCAAAAGAAAATGGCG
	R: CCTGCAAACAATTCACGAAT
*Caspase-8*	F: GGAGATGGACAGGCGTTCTTTAC
	R: CGATACCGTCCTTGTGGAACTCT
*Cyclin dependent kinase 6*	F: TCATAGTCAGCCACTCCAGC
	R: TTTAGGTCCGTGAACAACATC
*DNA replication licensing factor mcm 5*	F: GTAGCCGTTGGAATCCGTAA
	R: CGATGGTTTCATAGACATTTGG

## Data Availability

All study data are included in the article.

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
