# Peer review of "TMT-Based Quantitative Proteomic Analysis Reveals the Key Role of Cell Proliferation and Apoptosis in Intestine Regeneration of Apostichopus japonicus"

_ijms, 2024, doi:10.3390/ijms25084250_

Round 1
Reviewer 1 Report
Comments and Suggestions for Authors
Review report ijms-2906252:
The manuscript entitled “TMT-Based quantitative proteomic analysis reveals the key role of cell proliferation and apoptosis in intestine regeneration of Apostichopus japonicus” proposes a tandem mass tag (TMT)-based quantitation to investigate protein profile changes during the process of intestine regeneration by the means of nanoESI-UHPlC-Q-Orbitrap-MS and following Mascot analysis and cluster analysis and using sea cucumbers as animal model. Differential proteins involved in cell proliferation and apoptosis were identified and further validated using flow cytometry assay and Real-Time PCR.
The manuscript length is appropriate, and it is organized in a logical way.
The abstract is quite clear and sufficiently reflects the manuscript content. Similarly, the entire manuscript content is well structured and data analysis is very clearly presented.
The investigation on regenerative processes and their pathways are a relevant subject, so this type of study should be implemented. Some studies on intestine regeneration in echinoderms and, in particular Apostichopus japonicus, were published yet. However, in this study the usage of proteomics- based protocols, which is reported in few researches, is of great importance especially for the assessment of regenerative pathways.
The further confirmation of the activation of cell proliferation and apoptosis using Real Time-PCR and FACS validated the proteomics-based analysis. An added value compared to the literature is an excellent statistical analysis and a very good presentation of data through very well-presented figures and graphs.
I have some comments and suggestion for the authors.
Key issues:
(1) Materials and methods section should be implemented.
- The description of first part and of protein extraction and digestion should be detailed.
- Add (Company, City, Country) for all instrumentations reagents standards, software. Standardize and change it in all manuscript.
- Similarly for the LC-Q-Orbitrap-MS analysis only the chromatographic part of the method was detailed, while no details on the spectrometric part (or references of protocols used) was provided.
- How many replicates for each experiment? Specify it and, if necessary, add descriptive statistics used for evaluation of replicates.
- Do the same for section 2.5
- Add a brief description of flow cytometry protocol. This part is totally missing in the manuscript!!!
Please clarify and add these points.
(2) Discussion: I very appreciated in the discussion the comparison with previous works both of other authors and of this ms authors, however I suggest authors to better argument this part (from line 341 to line 360).
(3) Possible additional data: what is really missing in the manuscript is further confirmation of the activation of apoptosis (caspase-6 activation...) using western blot analysis. do the authors have the possibility to add this data?
(4) Conclusion: Conclusions section is missing. This section if essential and should be added.
I suggest authors add this part: it should give first again your objective, then show your major and most important results and achievements, underlining also the future perspectives of this interesting study.
Minor issues:
Line 87: do you mean RIPA lysis buffer? Better define it also the acronym and add(Company, City, Country)
BCA: define the acronym.
Attention to the mis-use of Saxon genitive
Line 103 FA: formic acid? ACN: acetonitrile?
Please control again all the acronyms.
Line 144: were
Line 254:were
Line 266: were
I suggest the authors to re-read carefully the manuscript and correct all typos and grammatical errors.
Based on these comments I strongly encourage the authors to improve the manuscript, since it is a suitable candidate for publication in International Journal of Molecular Sciences.

Comments on the Quality of English LanguageI suggest the authors to re-read carefully the manuscript and correct all typos and grammatical errors.
Author Response
-Reviewer 1
1. The description of first part and of protein extraction and digestion should be detailed.
Response: Thanks for reviewer’s sincere advice. We have provided a detailed description of first part and protein extraction and digestion part (Line 370-418).
2. Add (Company, City, Country) for all instrumentations reagents standards, software. Standardize and change it in all manuscript.
Response: Thanks for reviewer’s kind advice. We have supplemented the detailed information (Company, City, Country) of all instruments and reagents in the revised manuscript.
3. Similarly for the LC-Q-Orbitrap-MS analysis only the chromatographic part of the method was detailed, while no details on the spectrometric part (or references of protocols used) was provided.
Response: We appreciate the reviewer’s comment and we have added a detail description about the LC-MS/MS analysis (Line 444-460).
4. How many replicates for each experiment? Specify it and, if necessary, add descriptive statistics used for evaluation of replicates. Do the same for section 2.5.
Response: We appreciate the reviewer’s comments. In the TMT-Based quantitative proteomic analysis, the parallel sequencing groups were not included, and instead of increasing the sampling time points during different stages of intestinal regeneration in A. japonicus to monitor changes in protein expression. To ensure sample stability and comparability, we extracted proteins from a mixture of tissues from nine A. japonicus for each regenerative stage's protein sequencing analysis. Moreover, additional validation experiments such as qRT-PCR and flow cytometry were conducted to assess the reliability of this sequencing results. In the results, the detected levels of cell proliferation and apoptosis were consistent with the protein expression trends obtained through proteomics data, reflecting the reliability and effectiveness of the sequencing results. The statistics information in section 2.5 have been descripted in Line 505.
5. Add a brief description of flow cytometry protocol. This part is totally missing in the manuscript!
Response: Thanks for reviewer’s kindness minder, we have supplied the detail procedure for flow cytometry (Line 529-554).
6. Discussion: I very appreciated in the discussion the comparison with previous works both of other authors and of this ms authors, however I suggest authors to better argument this part (from line 341 to line 360).
Response: Thanks for the reviewer’s professional and valuable comments. We have supplemented the argument about the comparison with previous works both of other authors and of this study (Line 263-290).
7. Possible additional data: what is really missing in the manuscript is further confirmation of the activation of apoptosis (caspase-6 activation...) using western blot analysis. do the authors have the possibility to add this data?
Response: Thanks for the reviewer’s sincere and professional comments. We fully agree with the reviewer's viewpoint that supplementing protein imprinting data could further demonstrate the activation of cell apoptosis. However, due to the lack of specific antibodies, it is currently difficult for us to verify this result from a protein perspective. To further demonstrate this result, we selected several genes related to cell proliferation and apoptosis from Figure 5 A and Figure 5 E for qRT-PCR experiment. The results are shown in the Figure 5 D and Figure 5 H.
8. Conclusion: Conclusions section is missing. This section if essential and should be added. I suggest authors add this part: it should give first again your objective, then show your major and most important results and achievements, underlining also the future perspectives of this interesting study.
Response: We agree with the reviewer’s comment and added the conclusion section in the revised manuscript (Line 563-576).
Minor issues:
9. Line 87: do you mean RIPA lysis buffer? Better define it also the acronym and add (Company, City, Country). BCA: define the acronym.
Response: We appreciate the reviewer’s comment and we have added a detail description about these abbreviations (Line 374) and corrected similar issues in this manuscript.
10. Attention to the mis-use of Saxon genitive
Response: Thanks for reviewer’s nice minder, we have checked the Saxon genitive in the revised manuscript.
11. Line 103 FA: formic acid? ACN: acetonitrile? Please control again all the acronyms.
Response: We appreciate the reviewer’s comment and the abbreviation information for FA and ACN (Line 437-438) and similar issues throughout the manuscript have been completed in the revised manuscript.
12. Line 144: were; Line 254:were; Line 266: were
Response: Thanks for reviewer’s nice minder. We have fixed these errors and checked through the whole manuscript.
13. I suggest the authors to re-read carefully the manuscript and correct all typos and grammatical errors.
Response: We appreciate the reviewer’s valuable comments and have carefully re-read this manuscript. In addition, a colleague who is skilled at English has carefully checked the paper to correct potential grammatical errors.
Reviewer 2 Report
Comments and Suggestions for Authors
In the manuscript entitled ”TMT-Based quantitative proteomic analysis reveals the key role of cell proliferation and apoptosis in intestine regeneration of Apostichopus japonicus” the Authors tried to reveal the proteins involved in intestine regeneration of Sea cucumbers. The study is very interesting for the readers since it may reveal the arrangement of some of the proteins important in regeneration process giving another data base for analysis. In my opinion the manuscript requires some modifications before being accepted for publication.
Lines 60-61 - the sentence seems not to be not finished.
Line 87 - What was the weight of frozen tissue taken for protein extraction? Present the RIPA buffer composition and the volume of RIPA buffer used. What protease inhibitor was used and what was its concentration?
Lines 92-96 – present more detailed protocol for TMT labelling and fractionation.
Line 136 – what was the weight of samples used, what was the volume of Trizol used?
Line 144 -how the sequences of primers were obtained (with what tool were they designed?; if they were cited give the refrence).
Line 154-what is the goal of this sentence?
Give more details about tools used for genes and proteins analysis that allowed to obtain and present data in Table 1, Figure 1, Figure 2. Figure 3, Figure 4, Figure 5A and 5D. How many experiments were performed (n) to present the data?
Figure 2 – the detailed information is almost not visible – pleas enlarge ach of the element of Figure with KEGG pathway enrichment analysis
Figure 4 - Gene ontology enrichment analysis of DEPs should be enlarged
What was the source for presentation of expression level of proteins in Figure 5A and 5D?
How many experiments were performed (n) to present the data?
I suggest to present more detailed discussion about the results since the general “interaction between cell proliferation and apoptosis” is quite well known.
What is the impact of presented results on tissue regeneration “management” in human?
Author Response
-Reviewer 2
1.Lines 60-61 - the sentence seems not to be not finished.
Response: We appreciate the reviewer’s kind reminder. The sentence (Line 60-61) has been modified as shown in Line 73-78.
2. Line 87 - What was the weight of frozen tissue taken for protein extraction? Present the RIPA buffer composition and the volume of RIPA buffer used. What protease inhibitor was used and what was its concentration?
Response: We appreciate the reviewer’s comment and have supplemented this detail information in the manuscript (Line 373-377). (1) 100 mg tissue samples derived from each group were taken for protein extraction. (2) We have supplemented the lysis buffer’ volume and component (50mM Tris (pH 7.4), 150mM NaCl, 1% TritonX-100, 1% sodium deoxycholate, 0.1% SDS, 2mM sodium pyrophosphate, 25mM β-glycerophosphate, 1mM EDTA and 1mM Na3VO4) in the manuscript. (3) The protease inhibitor phenylmethylsulfonyl fluoride (PMSF) (Merck, Germany) at a final concentration of 1 mM used in this paper.
3. Lines 92-96 – present more detailed protocol for TMT labelling and fractionation.
Response: Thanks for the reviewer’s professional and valuable comments. We have supplemented more detailed protocol for TMT labelling and fractionation in the revised manuscript (Line 419-443).
4. Line 136 – what was the weight of samples used, what was the volume of Trizol used?
Response: We appreciate the reviewer’s comments. In this experiment, 50 mg of tissue was used to extract RNA, and 1mL of Trizol was added during the process. We have made revisions to the manuscript.
5. Line 144 -how the sequences of primers were obtained (with what tool were they designed?; if they were cited give the refrence).
Response: Thanks for the reviewer’s sincere and professional comments. The software used for designing primers and related references have been added to the manuscript (Line 512-514).
6. Line 154-what is the goal of this sentence?
Response: “Overview of changes in differential proteins between normal and regenerative stages .” This sentence is the subheading of result 3.1, and its format was not recognized when submitting the manuscript, so the purpose of this sentence in the article is unclear. Thanks for the reviewer’s kind reminder and we have made corrections to this error in the manuscript Line 93-94.
7. Give more details about tools used for genes and proteins analysis that allowed to obtain and present data in Table 1, Figure 1, Figure 2. Figure 3, Figure 4, Figure 5A and 5D. How many experiments were performed (n) to present the data?
Response: We appreciate the reviewer’s suggestion. The tools used for the genes and proteins analysis were supplemented in the revised manuscript Line 488-501. In the TMT-Based quantitative proteomic analysis, we did not set up parallel sequencing groups, but instead attempts to increase the sampling time points during different stages of intestinal regeneration in A. japonicus to monitor changes in protein expression. To ensure sample stability and comparability, we extracted proteins from a mixture of tissues from nine A. japonicus for each regenerative stage's protein sequencing analysis. Moreover, additional validation experiments such as qRT-PCR and flow cytometry were conducted to assess the reliability of this sequencing results. In the results, the detected levels of cell proliferation and apoptosis were consistent with the protein expression trends obtained through proteomics data, reflecting the reliability and effectiveness of the sequencing results.
8. Figure 2 – the detailed information is almost not visible – pleas enlarge ach of the element of Figure with KEGG pathway enrichment analysis
Response: We agree with the reviewer’s comment and add more clear and appropriately sized images in the revised manuscript.
9. Figure 4 - Gene ontology enrichment analysis of DEPs should be enlarged
Response: Thanks for the reviewer’s kind reminder. We have provided more clear and appropriately sized images in the revised manuscript. Due to the large pixel size of the image, the image quality in the manuscript has been compressed, resulting in an unclear appearance. In this revision, we have uploaded high-quality original images separately.
10. What was the source for presentation of expression level of proteins in Figure 5A and 5D? How many experiments were performed (n) to present the data?
Response: We have established a database of differentially expressed proteins at various time points (2-, 7-, 12-, 20, and 28-dpe group) during the intestine regeneration process in A. japonicus. Then, we combined literature reports and KEGG annotations to select pathways and proteins related to cell proliferation and apoptosis mechanisms. After further removing undefined hypothetical proteins, we used heat maps to draw the expression trends of each protein at different times during regeneration, eventually obtaining the results presented in Figure 5A and 5D. We have supplemented detailed analysis steps for this part of the data in the experimental method section (Line 493-501). For each regenerative group's sequencing experiment, we collected and mixed tissue from 9 A. japonicus individuals. Although the experiment did not have a parallel group for sequencing, subsequent q-PCR experiments and flow cytometry results on cell proliferation and apoptosis were consistent with trends in protein expression, indicating that the sequencing data had a certain degree of reliability.
11. I suggest to present more detailed discussion about the results since the general “interaction between cell proliferation and apoptosis” is quite well known. What is the impact of presented results on tissue regeneration “management” in human?
Response: Thanks for the reviewer’s sincere and professional comments. We have added some research progress on the interaction between cell proliferation and apoptosis, as well as a discussion on the impact of related studies on the "management" of human tissue regeneration (Line 326-358).
Round 2
Reviewer 2 Report
Comments and Suggestions for Authors
I have read the Authors' response with a huge interest - they have answerd some of my questions and improved the manuscript. However, I suggest to present the 4.2 not as an instruction, but as a description.
Already, I suggest the minor revision of the manuscript beacuse of mantioned above text edition.
Author Response
1. I have read the Authors' response with a huge interest - they have answerd some of my questions and improved the manuscript. However, I suggest to present the 4.2 not as an instruction, but as a description.
Response: Thanks for the reviewer’s professional and valuable comments. We have made modifications to section 4.2 according to the reviewer's comments, and the modified parts are highlighted in yellow (Line 371-421).
We have done our best to improve our manuscript and made changes wherever necessary. These changes do not influence the content and framework of our paper. We thank the Editors/Reviewers for their time and hope that the revised manuscript can meet with your approval.
Looking forward to your favorable decision.
Best wishes,
Sincerely yours,
Chenghua Li